# Access to Nature in a Post Covid-19 World: Opportunities for Green Infrastructure Financing, Distribution and Equitability in Urban Planning

**DOI:** 10.3390/ijerph18041527

**Published:** 2021-02-05

**Authors:** Ian Mell, Meredith Whitten

**Affiliations:** 1Department of Planning & Environmental Management, School of Environment, Education & Development, University of Manchester, Manchester M13 9PL, UK; 2Department of Geography and Environment, London School of Economics and Political Science, London WC2A 2AE, UK; m.whitten@lse.ac.uk

**Keywords:** green infrastructure, urban planning, health, finance, equity, accessibility

## Abstract

Covid-19 changed the way many people viewed and interacted with the natural environment. In the UK, a series of national lockdowns limited the number of places that individuals could use to support their mental and physical health. Parks, gardens, canals and other “green infrastructure” (GI) resources remained open and were repositioned as “essential infrastructure” supporting well-being. However, the quality, functionality and location of GI in urban areas illustrated a disparity in distribution that meant that in many cases communities with higher ethnic diversity, lower income and greater health inequality suffered from insufficient access. This paper provides commentary on these issues, reflecting on how planners, urban designers and environmental organizations are positioning GI in decision-making to address inequality. Through a discussion of access and quality in an era of austerity funding, this paper proposes potential pathways to equitable environmental planning that address historical and contemporary disenfranchisement with the natural environment in urban areas.

## 1. Introduction

The history of urban planning has been structured by significant intersections with public health, discussions focusing on the equitable distribution of public resources and the difficulties faced by governing bodies in ensuring that all members of society are treated fairly. Ebenezer Howard’s Garden City principles were, in part, a reaction to the squalor and urban health crises of the 1800s [1,2]. Moreover, in New York, Frederick Law Olmsted’s landscape designs were created to address growing health inequalities [3,4]. However, the extent to which health, well-being and social equity structure fairer and more livable cities varies. Although we can identify examples of the successful integration of people, place and nature [5,6], there is extensive evidence from Europe, North America and South and East Asia of cities expanding with little reference to the development of interactive, high-quality or functional places [7].

The assessment of whether cities are responding effectively to the needs of their communities, their economies and their environment was thrown into sharp relief in 2020 by the Covid-19 pandemic. In the UK, for example, people had their freedom of movement curtailed via a series of national lockdowns in 2020 and 2021. Parks and other public open spaces, however, remained accessible and were promoted by the UK government as essential infrastructure enabling people to engage in a daily allowance of outdoor activity [8]. Some areas chose to close their parks due to safety fears, only to receive negative publicity. Other countries, including Spain and Italy, imposed more stringent lockdowns, limiting all but essential engagement with spaces outside the home, i.e., trips to grocery stores [9,10].

The variability by which green spaces, known collectively in academic and practice-based work as green infrastructure (GI), have been approached by policymakers has been discussed extensively during the pandemic, providing a lens to assess how government, communities and other organizations within the landscape, development and planning sectors consider “nature” as part of urban development/management conversations.

In this paper, GI is defined as a “natural life support system—an interconnected network of waterways, wetlands, woodlands, wildlife habitats and other natural areas; greenways, parks and other conservation lands; working farms, ranches and forests; and wilderness and other open spaces that support native species, maintain natural ecological processes, sustain air and water resources and contribute to the health and quality of life for America’s communities and people” [11], and refers to the matrix of green and blue spaces within and across urban/rural areas that provide socio-economic and ecological benefits to a variety of communities of interest.

Throughout the GI literature there is an ongoing debate which (a) attempts to characterize GI with specific thematic, spatial and scalar classifications and (b) asks whether this progresses GI thinking effectively. In this paper we do not propose to further this debate. The variation in the types, scales and benefits associated with GI can be seen in Table 1 below, although this list is not exhaustive and is subject to variation depending on location and geographical context. Table 1 highlights the breadth of discussion surrounding what GI is, what spatial form it takes and what benefits it delivers. It also promotes the multifunctional aspect of GI in delivering health and well-being, biodiversity, climate change and social inclusivity benefits, as outlined by Benedict and McMahon in their seminal text [11].

In this paper we discuss access, equitability and the financing of GI predominantly at the local, neighborhood and city scale. This means we engage more frequently with GI as urban parks, greenspace, waterways and street trees as these are public GIs that are influenced by issues of accessibility and functionality. Presenting the paper in this way provides a platform arguing that GI should continue to be considered as “essential infrastructure”. It is also reflective of the scales/GI types discussed by Nesbitt et al. [6], Public Health England [8], Rousseau and Deschacht [9], Ugolini et al. [10] and Benedict and McMahon [11] within urban planning debates in the transition to a post-Covid-19 landscape, as well as by the Prime Minster of the UK and others during the pandemic.

The aim of this paper is to provide a commentary—and not a set of novel methodologies, approaches or research—situating the role of GI within Covid-19 debates. Lennon [12] articulated this view, noting that issues of form and features, distribution, connectivity and resilience within planning are of paramount importance to our responses to Covid-19. Rastandeh and Jarchow [13] go further, outlining a program of action that places nature and enhanced ecological resources at the center of change. Both papers argue that issues of distribution and equity in spatial planning are complex and that any reaction to Covid-19 needs to take into account the scope of the ecological, economic and social factors influencing urban functionality and health.

Our paper therefore asks how the issues of accessibility, equitability in terms of the distribution of GI and the ongoing complexity of funding have influenced the discussion of landscape value and functionality before, during and potentially after the Covid-19 pandemic. This provides insights into the relationship between urban planning, environmental management and public uses of GI, identifying where good and poor practice exists.

We do not explicitly examine the well-established links between health and GI during Covid-19 but alternatively use the pandemic as a backdrop to discuss a series of factors which influence the use of green spaces, and therefore have a significant role in promoting health and well-being. This is a purposeful decision. A wealth of evidence supports the links between health and GI [14,15] but there is little debate focusing on the implications of accessibility related to Covid-19. Moreover, throughout 2020 academics and practitioners have extensively debated changes in health due to Covid-19 and we do not seek to replicate this discussion. Consequently, we provide more directed commentary on the ongoing issue of the equitable provision of GI in an era of financial uncertainty. We do this from a UK perspective, using examples of policy, practice and financing to support our discussion. This is supplemented with additional commentary from outside of the UK to illustrate that these are not geographically isolated issues only relevant to the UK.

By examining the academic and practitioner research focusing on distribution, equitability and access within an analysis of the impacts of austerity (in the UK and internationally), the following examines the perceived disparities between how GI is delivered and how Covid-19 has led commentators in built and natural environment disciplines to reconsider its value to society. The financial component to this discussion is important as its signposts current government thinking regarding the added value of “nature” to society and reflects which communities have historically benefitted from provision. The lack of a baseline economic position for GI remains problematic, raising concerns within government of whether nature is cost-effective or indeed needed to support a prosperous society [16].

The commentary concludes by looking at the opportunities available to decision-makers to ensure that health, well-being and livability are used to structure development that aids public health in the UK. We also feel that a note of clarification is needed. This paper does not offer the “solution” to the spatial inequality of GI provision and we do not introduce new empirical work to support the recommendations for change in policy or practice. Instead, the paper uses evidence to analyze a series of existing issues that have been exacerbated by the Covid-19 pandemic and thus provide commentary which should form the basis of urban development discussions going forward.

## 2. Understanding the Value of Green Infrastructure in a Covid-19 Landscape

The academic and practitioner literature focusing on GI illustrates a myriad understanding of the concept’s principles, applications in practice and benefits to a variety of stakeholders [17,18]. Although a level of consensus has developed within local government and the environment sector supporting investment in GI as a series of ecological networks connecting people to the landscape within and across urban areas, whilst also delivering multifunctional socioeconomic and ecological benefits, there is continued variability in the ways in which this knowledge is applied. A key aspect of diversity is the way in which GI is aligned with key policy areas in different locations. For many commentators, GI is a core component of the health and well-being agenda [19]; for others it reflects an approach to stormwater management [20] whilst also responding to changes in biodiversity and climate change [21]. Moreover, GI has been linked with economic development narratives, real estate and the politics of environmental decision-making [22,23,24]. The outcome of these varied approaches is a complex interplay of investment agendas, thematic uses of the concept and delivery located within a combined strategic/local context. Collectively, this means that the “value” of GI (conceptually, practically or from social, economic, ecological and political perspectives) varies extensively between locations and the subsequent appreciation of “nature” within urban planning may lack transferability between stakeholders.

As the impacts of Covid-19 became increasingly prominent in the UK, and internationally, the academy commenced on a process of reflection focusing on the influence of both the physical form and use of the built and natural environment as part of societal reactions to the novel coronavirus. For example, Town Planning Review commissioned a series of viewpoints debating reactions to Covid-19 in development, livability and urban governance [25]. Within these debates, significant attention centered on the mental and physical health impacts of Covid-19 and the specific role that urban nature played as a response [26]. However, they also focused discussions on the ways in which GI as a set of places, a set of amenities and a set of processes and functions is utilized in planning. This reinforced our understanding of the variation in approach to GI delivery and furthermore the diversity of attention afforded to the natural environment in planning praxis [27].

To better understand the lack of consensus regarding the value of GI to society, we must unpack the socioeconomic and political rationales of environmental management, especially in urban areas where access to nature is potentially more difficult compared to rural areas. This moves beyond simple interpretations of policy as being either pro-economic development or pro-ecological resilience to a more nuanced appreciation of the decision-making process within government—i.e., policy-making, practice and law—and the value it places on GI, as part of a diversifying suite of resources supporting investment by developers, the environment sector and local government [28,29]. Illustrating this point, Walmsley [30] noted that many built environment specialists consider urban greening as “something nice to have” once investment in built infrastructure had been completed. However, we can identify a significant shift in emphasis away from this view, via the evidence produced by EU-led projects including the Green Surge [31], Valuing Attractive Landscapes in the Urban Economy (VALUE) [32] and current Horizon 2020 Nature-Based Solutions (NBS) [33] projects, to a narrative of investment in nature considered as a positive.

Unfortunately, support for GI has not been forthcoming in all areas of government in the UK or internationally. In many instances, this reflects the difficulty in ascribing economic value to nature [34]. Research, however, argues that there is an explicit value of nature to society. Researchers including Tyrväinen [35], Qureshi, Breuste and Lindley [15], Dempsey, Smith and Burton [36], Cilliers, Cilliers and Juaneé [37] and Mell [38] have all noted the added socioeconomic value delivered via increased investment in nature. The Covid-19 pandemic has therefore strengthened the call for government to engage more effectively with this evidence.

Moreover, Public Health England [8,26] has argued that GI is a crucial component of the UK’s response to Covid-19 and of attempts to improve public health more generally. The evidence released underpinning their Improving access to greenspace: A new review for 2020 [26] report suggested that savings of £2.1 billion per year could be made in health costs if everyone in England had access to high-quality greenspace. This figure was linked to a proposed increase in physical activity associated with such spaces. They also reported that:in Birmingham, the annual net benefit to society of the city’s parks and greenspace is nearly £600 million, which includes £192 million in health benefits;in Sheffield, for every £1 spent on maintaining parks, there is a benefit of £34 in health costs saved, with local residents being the primary beneficiaries;in England and Wales, houses and flats within 100 m of public greenspace are an average of £2500 more expensive than they would be if they were more than 500 m away—an average premium of 1.1% in 2016, suggesting that the public places a value on being near to greenspace.

As a consequence, Covid-19 increased the calls for additional funding to be released from central to local government to fund GI. Commentators also argued that this might facilitate a reconsideration by central government of the value of nature and promote engagement with the evidence base supporting the economic, sociocultural and ecological benefits of GI [7,39,40].

To date, this process has included a broad response from within UK government. We have also seen comparable responses from local government in US, Japanese, Australian and other EU cities, recognizing that, although variation in size, function and amenity are prevalent in GI planning, it provides essential locations for sports, recreation and health improvements [41,42,43,44]. Furthermore, this framing of value could be considered as a representation of Mell’s [45] notion that all GI is valuable, as it provides a cumulative benefit to society that is not restricted by size, location or quality. A review of the academic literature supports this view, highlighting the role that GI holds in facilitating social interaction, play, community bonding and well-being [14,46,47], each of which are not dependent on an explicit scale or set of amenities to be delivered in all locations.

As lockdown measures diversified across the UK, there has been a corresponding increase in the attention placed on GI in government, in terms of access to high-quality spaces, and from a public health perspective. These discussions have, in many instances, focused on the location, provision and utility of GI within urban areas and whether all members of society have access to the same types and quality of spaces. They have also reflected the growing primacy called for by academics and practitioners for GI to be considered as “essential infrastructure”, a proposal historically undermined by built infrastructure and economically centric organizations [38]. The following sections reflect on these issues using practice in the UK to structure debate but with supporting evidence from other geographical locations.

## 3. Financing Green Infrastructure

The UK government’s positioning of GI, in the form of public parks, commons, gardens and other ecologically focused places, e.g., nature reserves, as critical infrastructure during the pandemic raises significant questions about their decade-long underfunding of landscape management via austerity measures. Since 2010, successive UK governments have enacted significant cuts on local government and environment sector funding, leading local planning authorities (LPAs) to cut between 20 and 90% of their budgets for environmental management. Further, LPAs’ budgets for discretionary services, under which greenspace delivery and management typically falls, have experienced deep cuts [48]. The impact of this has been a contraction of the quality of landscape management and a curtailing of strategic development in new resources due to a lack of capital investment and revenue funding. Mell’s [24] discussion of this process, and the subsequent analysis of its limitations, as well as the opportunities for innovation developed in Liverpool [49], illustrate the complexity of balancing budgets with service provision (see also Table 2). Furthermore, the precarious nature of LPA financing has led to significant conflict within LPAs as they continue to debate how to deliver legally required services, i.e., education, whilst retaining funding for supplementary services such as GI.

Further complications were identified by the Greater London Authority (GLA) [40] in their analysis of the financial impacts of Covid-19 on parks. These include loss of income and increased maintenance costs associated with increased patronage and the activities they engage in. In London, the GLA identified the loss of income from users, concessions, car parking and funeral services, i.e., crematoria, as significantly impacting revenue. Further examples from Leeds, Nottingham, Plymouth, Rugby, Walsall and Watford illustrate similar concerns. Losses are forecast to be between 5 and 86%, with an average loss of 39% [40]. The GLA also argued that the long-term viability of LPA funding for GI has been compromised, and as a result alternative sources of funding need to be identified.

Covid-19 thus provides scope to consolidate the ongoing debates regarding the funding of GI in the UK, focusing political attention on the value of nature to society. Furthermore, within the academic and practitioner literature, a corresponding discussion exists focusing on what type of GI is being funded and where funding is being allocated (geographically across the UK and within individual LPAs). Table 2 presents these issues using a traffic light system—with red indicating a lack of sustainable funding sources, amber indicating some sustainable funding sources and green indicating that sustainable sources of funding have been identified—to visually represent a synthesis of reported discussions of GI funding. Where types of space are reported as “red”, this is reflective of a lack of permanence of funding being reported by LPAs and other GI advocates. Where categories are “green”, and therefore sustainable, these reflect the growing number of land management agreements attached to private development or public–private investment [24,49,50,51,52].

A reading of Table 2 suggests that there is a lack of sustainable funding available to support investment in larger or public GI resources. This is supported by practitioner discussions in the UK who view the ongoing cuts in government funding as systematically undermining their ability to deliver effective environmental management at site, neighborhood and city scale. However, evidence presented by Mell [24,49] suggests that there is a growing appreciation within the business community, non-environmental organizations and the construction industry that investing in GI is beneficial both economically and in terms of reputation, with organizations continuing to focus their work on project/site-based intervention. Moreover, within LPAs we can identify an evolving understanding of how the ecosystem and sociocultural values associated with nature at a larger scale can be integrated into development plans or strategies, even where corresponding funding has not been allocated. A note of caution is needed, however, as the wider concerns of equitable access may not be addressed through the implementation of private and discreet development. This raises questions over whether a guerilla or informal process of urban greening could lead to greater access to or proportions of urban greenspace. The process of rewilding or community use of unoccupied/underused spaces in urban areas has been championed by Incredible Edible in the UK as a way of maximizing the use of public space without the constraints of ownership; alleyway greening in Liverpool can be added as a similar practice [53]. To date the uptake of these activities remains piecemeal but could circumnavigate the needs for the allocation of formal funding streams for GI.

## 4. Equitability of Access to Green Infrastructure

A series of reports released in 2019/20 examined the equitability of GI in cities in the UK [26]. Focusing on discussions of quality, quantity and functionality, these debates were framed as an analysis of the composition of urban form. We can also question whether access to GI is the same as having access to the benefits associated with green space. This distinction refers to the equitability of physical access and distribution of resources to all members of society and the ways in which people can benefit from a reduction in air pollution, flood protection and health improvements associated with GI provision. The former does not explicitly lead to the latter and is an issue that requires consideration in discussions of equitability.

However, a growing evidence base running in parallel to this analysis explicitly reflects on the political decision-making process, questioning why low-income and Black, Asian and Minority Ethnic (BAME)/Black, Indigenous and people of color (BIPOC) communities are subject to significantly lower provision, as well as poorer quality spaces and management. These reports are geographically diverse in terms of the cities/countries they discuss but support the analyses of Safransky [54], Anguelovski et al. [55] and Nesbitt et al. [6], who argued that there has been a systematic exclusion of high-quality GI provision in communities that are home to predominately BAME and/or low-income residences. Their work highlights a critique of investment in urban nature regarding the equitability of distribution. Engaging with international evidence is useful in situating the current discussions in the UK, as it illustrates that these issues are not isolated to UK contexts. The studies by Anguelovski et al. and Nesbitt et al. therefore provide important comparators to help shape practice.

Within a UK context, Natural England’s People and Nature Survey for 2020 [56] reported significant inequality in access to and use of natural spaces, noting that 71% of children from BAME families spent less time outside/in a greenspace during lockdown compared to 57% of white children. They also found that children in lower-income families spend less time outside than other children (73% compared to 57%). The marginalization of people due to income and ethnicity has also been linked to housing, with these social groups more likely to live in smaller homes in higher-density locations [57,58]. As a consequence, there are increased problems associated with the limited availability of public or private greenspace, as discussed by the Office for National Statistics (ONS) [59], who state that 12% of UK residents have no garden (21% in London). Economically deprived neighborhoods also have fewer accessible, high-quality public green spaces, and residents in these areas experience poorer health than those who live in green environments. Furthermore, communities identifying as Black in the ONS survey are four times less likely to have access to outdoor/private space compared to those identifying as White.

Disparities in access to greenspace have been linked to long-term physical and mental health issues. However, research shows that health inequalities are halved in greener areas. Covid-19 has, therefore, exacerbated the impact of a lack of access to GI for specific groups of society, raising questions for planners regarding current and future provision. This may be a byproduct of broader structural inequalities that exist within planning, and could be considered as an outcome of a targeted provision of poorer quality and/or a decreased proportion of greenspace in areas with high ethnic diversity or more numerous low-income families. The studies by Agyeman et al. [60], Anguelovski et al. [55] and Shi [61] discussed these issues internationally, and throughout 2020 these critiques (and others) have become increasingly prominent in a UK context. Illustrating this disparity is the fact that the wealthiest areas in London have approximately 10% more public space compared to its most deprived areas [62]. Furthermore, communities with higher proportions of BAME/low-income residents receive proportionally less spending per person on GI [63]. Communities living in high-density areas with limited GI provision are as a consequence overtly constrained in how they engage with nature for exercise or socially distanced interaction. Analysis also found that the 20 poorest LPAs in the UK experienced a 28% reduction in use of greenspace during the pandemic/lockdown, whereas the 20 wealthiest areas showed no change [64].

Moreover, in London an increase in overall visits to local parks and green spaces was driven largely by younger and wealthier residents [65]. Such changes are compounded due to the contrasts in personal mobility between communities of low and high income, with the latter being able to travel to access spaces of higher quality (due to affluence, proximity or access to a private vehicle). The impacts of this disparity include worse reported mental and physical health, increased instances of anxiety and stress, conflict within family units, and a subsequent decrease in engagement with GI. It may also suggest that, even where access is afforded, the wider benefits afforded through interaction with GI may not be fully realized by all communities. If the quality of those spaces is variable they may not deliver the breadth of ecosystem and socioeconomic services required to support personal or communal well-being.

Furthermore, in August 2020 the UK government announced potential planning reforms that would allow deregulation of development, permitting conversion of office to residential use without needing planning permission. One significant consequence of this is the bypassing of requirements for the provision of access to open/amenity/green spaces to support additional populations [66]. Only 3.5% of such permitted development units, most frequently located in commercial and industrial areas with little public greenspace, have access to private amenity space, compared to 23.1% of housing units delivered through planning permission [66].

Whilst these issues are not solely the provision of BAME/BIPOC or low-income communities, they are reported as occurring more frequently in these locations than in high-income areas. We can also argue that a generational inequity exists, with younger urban residents less likely to have access to a garden and living in smaller (m^2^) living spaces than older people or those in rural areas [67,68], thus increasing their reliance on the availability of nearby and accessible green spaces.

One area somewhat absent from these debates is the role of local communities in communicating their needs to decision-makers to aid the focus, design and implementation of new GI. Whilst extensive evidence exists examining the positive impact that communities have on the resilience of urban greening projects in meeting local needs, there remains a disconnect in many places between what is required and what is planned. As a consequence of Covid-19, we may see a greater level of engagement with communities, especially those in high-density areas, as planners attempt to address issues of quality, quantity and distribution within local planning and development frameworks.

## 5. Distribution of Green Infrastructure in Urban Areas

The distribution of GI also varies due to the specificities of urban form, historical decision-making, perceived need and, in a number of cases, demographic diversity [69,70]. Furthermore, due to higher densities, urban areas have less significant proportions of GI (m^2^ or overall proportion (%) of greenspace) compared to suburban or rural areas. However, a growing literature argues that the size of a GI resource does not necessarily equate to its societal value [45,53]. Thus, smaller spaces that service neighborhoods can be of equal importance as a city-scale park in terms of accessibility or functionality [46]. As a consequence, the value of smaller GI sites can be considered to hold a greater cumulative impact on local use because they provide a more discreet level of infrastructure that is proximate and accessible [71]. Research has also shown that people are more likely to engage regularly with green spaces near where they live than with larger “destination” spaces [72,73,74]. However, the breadth of GI types, as noted in Table 1, provide scope for people to engage with larger sites for specific activities, e.g., sports, walking or cycling. Larger sites, e.g., country parks, therefore remain important as they have a greater capacity to support multiple users and uses simultaneously without showing signs of overuse. This has been a critical factor in facilitating community use of larger sites in 2020, as it has enabled people to maintain social distancing. Furthermore, as Mell [45] discussed, we need to be conscious of the complementary value of differently sized resources, as they support the development of a network of spaces that can service a wider range of socioeconomic and ecological needs when compared to a single type of GI.

In addition, although accessibility, functionality and amenity value are important facilitators of use at the local scale, we can also argue that the microclimatic benefits of temperature moderation delivered via shade, tree canopy and vegetation are critical to community use, as they improve the “comfort” of that location [75]. We also need to be aware that, within urban environments, the visible level of greenery retains an importance, along with the distance and routes to green spaces. This has a significant impact on how people value GI in urban areas in terms of providing a contrast to other built infrastructure [76].

An understanding of urban form is critical if we are to appreciate whether access to GI is equitable. In a significant number of cities in the UK, we can identify cycles of development, decline and regeneration that have reshaped the physical landscape [77]. Examples exist where the perceived need to diversify housing types to increase density and the provision of transport and commercial/industrial infrastructure have modified physical form at the expense of GI. Furthermore, as competition for space has increased, the conversion of GI in the form of public parks, water bodies and amenity green spaces into built infrastructure has increased. Evidence from the US [78] and Australia [79], as well as from the UK [67], argues that these processes have adversely impacted low-income and BAME/BIPOC communities most. Thus, urban areas including Liverpool, Belfast and London have been impacted twice: first by the removal of green spaces and their replacement with housing and roads bisecting neighborhoods; and second via a lack of appropriate replacements of green spaces [40,80].

In addition, there is also a growing analysis of whether metrics/indices associated with the size, distance and distribution of GI are valuable in assessing patterns of distribution and access. In the UK, the Accessible Natural Greenspace Standards (ANGSt) have shaped the thinking of planners regarding the provision of GI from the mid-1990s onwards [81]. ANGSt compliments the Fields in Trust “6 Acre standard” (and more recently the Green Spaces for Good 2018–2022 Strategy) and continues to shape investment in the UK. A selection of the most frequently discussed UK-based GI accessibility and amenity scoring systems are shown in Table 3, highlighting how size, time and distribution metrics are currently used to guide investment. It should be recognized though that additional standards from the US and Europe had have an influence on the evolution of these UK-based metrics and subsequently on assessments of accessibility.

Each benchmark/standard sets specific distance, travel time and size metrics, considered by many LPAs and construction industry organizations to denote the maximum requirements for GI provision. However, despite their value to central and city governments, critiques suggest that the fixed nature of metrics may actually limit their utility. It has also been suggested that cataloguing amenities or setting a 10-minute metric for the distance to the nearest green space fails to acknowledge variations in street hierarchies, personal mobility, site functionality (and its utility to users), the amenity value of a site or the wider spatial context of GI within an urban area [82]. As a consequence, although standardized metrics remain useful tools for policy-makers to assess provision, they are contested in terms of how effectively they shape delivery and whether they are appropriate mechanisms to meet the health, well-being and recreational needs of a given community. Greater flexibility may therefore be needed, i.e., the integration of a set of tolerances to contextualize provision in terms of size and distance, to facilitate a more nuanced understanding of quality and functionality.

## 6. Conclusions: An Approach for Effective Change

The Covid-19 pandemic illustrated beyond reasonable doubt the value of GI to government at all scales, as well as to the public. Although a reluctance to fund or equitably distribute GI provision in many cities remains, especially in those where land values for real estate development undermine the delivery of greenspace, there is a groundswell of support for investment in urban greening. To achieve this, a number of proactive steps need to be taken to position GI as “critical infrastructure”. This will enable city and metropolitan governments and developers to deliver provision that is equitable, of high quality to all and ensure that growth is embedded with environmental considerations. To achieve this, five areas can be identified as being fundamental to these changes.

### 6.1. Legislation

Ensuring equitability of GI requires a supportive legislative framework at a national and a local/LPA scale. Legislation requiring provision of GI in all new developments, with costs borne by developers, would increase the amount of greening and deliver improved benefits, particularly for health, climate change adaption, air quality and biodiversity. Use of tools such as the Urban Greening Factor, which has been adopted by the GLA and several London boroughs, could facilitate practical implementation. The development of legislation that explicitly requires the delivery of GI in low-income, high-density and ethnically diverse areas should be promoted as a policy priority to provide a framework to eliminate disparities in provision.

### 6.2. Funding

National government should prioritize/ring-fence funding for GI linked to health, well-being, climate change and equitability. Attaching financial support to provision via these policy areas would enable LPAs to make long-term commitments to GI funding, as it would align public health, environmental quality and economic development agendas into political support for investment. Guaranteed central government funding would also provide a baseline position for LPAs to attract match funding from corporate sponsorship, business and local taxes and community modes of funding to support GI development.

### 6.3. Equitability of Provision That Is Meaningful to All

Policy directives must commit to deliver GI for low-income and BAME/BIPOC communities as a priority. This should align mandatory delivery within national, regional and local/city-scale policy, drawing from existing metrics and demographic trends to identify areas of need. It should be independent of local political rhetoric and based on an understanding of societal change, a diversifying urban form, and the longer-term structure of an urban area. LPAs should develop transparent policy mandates that address historical inequality in provision and quality and direct investment to these locations in the first instance. Such policies should be underpinned with reference to existing funding mechanisms to ensure delivery.

### 6.4. Establishing an Economic Value for Green Infrastructure

Policy-makers need to engage more directly with the evidence attributing an economic value to GI and position investment in landscape enhancement centrally in discussions of gray/built infrastructure development. This requires consideration of how the benefits to health, climate change, water management and economic uplift associated with GI can be valorized to ensure it becomes an essential form of urban infrastructure. A wealth of evidence exists, drawn from Willingness-to-Pay (WTP), land capture and economic modeling analyses, illustrating the added value that GI generates for investment. By engaging with the range of techniques and the language of economic policy an additional process of rationalization can be integrated into strategic decision-making promoting long-term investment in environmental enhancement. The use of economic valuation should be considered a useful approach to wider assessments of GI value. However, there should be a corresponding call for greater attention to be paid to the ways in which ethnicity and affluence influence whether people can access, use and realize the benefits of GI.

### 6.5. Community-Led Planning for Equitable Green Infrastructure Provision

Although not explicitly discussed within this paper, there is a growing consensus that community-led action is needed to support planners, designers and the landscape profession in developing places that are accessible, meaningful and functional for local populations. It should therefore not be forgotten that communities are acutely aware of the barriers to the use of GI and can provide innovative solutions to size, funding and design constraints. However, although participatory planning and citizen science are gaining prominence, LPAs and other actors need to embrace local knowledge more readily. Integrating an understanding of the right project in the right place with the most appropriate mix of amenities is critical to improving access and supporting engagement with the breadth of benefits that GI affords. A bottom-up approach would potentially ensure that the resources being created are resilient to climatic, financial and sociodemographic change.

Each of these five areas provides a starting point for a more detailed discussion of how our cities should be managed in light of the impacts of Covid-19. However, although a robust evidence base exists supporting each of these areas, further academic and practice-based inquiry is needed to integrate the added value of investment of GI in the most appropriate policy, decision-making and development settings. All five statements should therefore be used to support the gathering of evidence and its integration into policy to promote more sustainable urban planning as the core form of investment. Issues of equitability, gentrification and funding related to economic analysis need to be taken into account in these debates. Our responses to Covid-19 have, as a consequence, facilitated a greater level of reflection on these issues, questioning the decisions being made in urban development. GI is one of the few resources that has increased its societal value during the pandemic, due to its function in providing locations for physical and social interactivity. The time is now right to push for GI to be considered as “essential infrastructure” and to legally embed its development within all policy to ensure that low-income or ethnically diverse communities do not continue to be excluded from the benefits of green space. This will take strong leadership, sustained funding and clear legislation but with political support cities can promote equitable access and the benefits of GI for all.

## Figures and Tables

**Table 1 ijerph-18-01527-t001:** Green infrastructure (GI) typologies.

Types of GI	Scale: Site (SI), Street (ST), Neighborhood (NE), City (CI), Landscape (LA)	Benefits	Site/ Corridor/Network
Street trees	SI, ST, NE, CI	Biodiversity enhancement, habitat creation, climate mitigation/microclimate moderation, interception of rainfall, places for economic development, location of social interaction, communal health and well-being	Corridor
Urban parks	NE, CI	Biodiversity enhancement, habitat creation, climate mitigation/microclimate moderation, interception of rainfall, location of social interaction/play, economic development opportunities, personal/communal health and well-being	Site
Private gardens	SI	Biodiversity enhancement, habitat creation, personal health and well-being	Site
Public gardens	SI, NE, CI	Biodiversity enhancement, habitat creation, climate mitigation/microclimate moderation, interception of rainfall, location of social interaction/play, economic development opportunities, personal/communal health and well-being	Site
Amenity greenspace	SI, NE	Biodiversity enhancement, habitat creation, climate mitigation/microclimate moderation	Site/ corridor
River corridors/fronts	NE, CI, LA	Sustainable transport, biodiversity enhancement, habitat creation, climate mitigation/microclimate moderation, location of social interaction/play, economic development opportunities, personal/communal health and well-being	Corridor
Lakes/ponds	SI, NE, CI	Biodiversity enhancement, habitat creation, climate mitigation/microclimate moderation, location of social interaction/play, economic development, economic development opportunities, personal/communal health and well-being	Site
Urban woodlands	SI, NE, CI	Biodiversity enhancement, habitat creation, climate mitigation/microclimate moderation, location of social interaction/play, economic development opportunities, personal/communal health and well-being	Site
Forest	CI, LA	Biodiversity enhancement, habitat creation, climate mitigation/microclimate moderation, economic development opportunities, personal/communal health and well-being	Site
Green walls/roofs	SI	Habitat creation, climate change mitigation, flood mitigation, urban cooling, reduced energy costs	Site
Play areas	SI, NE	Location of social interaction/play, economic development opportunities, personal/communal health and well-being	Site
Green cycle routes	NE, CI, LA	Sustainable transport, habitat creation	Corridor/network
Infrastructure greening (roadside greening)	NE, CI, LA	Habitat creation, aesthetic greening/screening, flood mitigation, climate change mitigation	Corridor/network
Sustainable drainage systems (SUDS)	SI, NE	Biodiversity enhancement, habitat creation, climate mitigation/microclimate moderation, interception of rainfall, economic development opportunities, personal/communal health and well-being, aesthetic improvements	Site/corridor
Allotments/urban agriculture	SI, NE, CI	Personal health and well-being, climate change mitigation	Site
Formal green belts	CI, LA	Habitat creation, climate change mitigation, sustainable transport, outdoor recreation,	Corridor/network
Pocket parks	SI, NE	Biodiversity enhancement, habitat creation, climate mitigation/microclimate moderation, location of social interaction/play, economic development opportunities, personal/communal health and well-being	Site

**Table 2 ijerph-18-01527-t002:** Financing green infrastructure.

Types of Space	Scale of Investment	Funder	Perceived Sustainability of Funding Source
Public parks	City, neighborhood	Local government, central government, private sponsorship, philanthropic gifts, S106, commercial revenue, CSR, community groups, EU funding/grants, environmental sector, charity funding, developers	
Public–private parks	Site, neighborhood	Private organizations, developers, local government, environmental organizations, community groups	
Street trees	Street, site	Local government, central government, private sponsorship, philanthropic gifts, S106, commercial revenue, CSR, community groups, EU funding/grants, environmental sector, developers	
Public gardens	Site, neighborhood	Local government, central government, philanthropic gifts, S106, community groups, EU funding/grants, environmental sector, charity funding	
Sports fields	Site, neighborhood	Local government, private sponsorship, commercial organizations, developers	
Riverside/riparian corridors/water bodies	City, regional	Local government, community groups, EU funding/grants, environmental sector, charity funding	
Urban woodland	Site, neighborhood, city	Local government, philanthropic gifts, S106, community groups, environmental sector, charity funding	
Amenity greenspace/grass verges	Site, neighborhood, city	Local government, infrastructure providers, developers	
Green walls/roofs	Site	Developers, business owners, environmental sector, local government, utilities providers	
Public plazas	Site, neighborhood	Private organizations, developers, local government, environmental organizations, community groups, local government	
Housing/commercial development greenspace	Site, neighborhood	Developers, business owners, environmental sector, local government, local community	

**Table 3 ijerph-18-01527-t003:** UK greenspace metrics.

Metric name	Organization	Scope
Accessible Natural Greenspace Standard (ANGSt)	English Nature/Natural England	Based on five categories: 1. No person should live more than 300 m from their nearest area of natural greenspace of at least 2 ha in size. 2. At least one accessible 20-ha site within 2 km of home. 3. One accessible 100-ha site within 5 km of home. 4. One accessible 500-ha site within 10 km of home. 5. Provision of at least 1 ha of Local Nature Reserve per 1000 people.
Guidance for Outdoor Sport and Play	Fields in Trust	Recommendation for a green space and/or playing field of at least six acres in size per 1000 head of population. An additional 10-min walking distance metric has been added more recently.
Building with Nature	Building with Nature/Gloucestershire Wildlife Trust	Accreditation based on four areas of thematic standards, based on three levels of accreditation: design award, full award (good), full award (excellent). 1. Core standards—(a) multifunctionality; (b) uses local landscape character features as starting point and respects/incorporates local context; (c) type, quality and function respond to local policy context; (d) GI features are resilient to Climate Change and minimize environmental impact of scheme; (e) provision is made for long-term management and the maintenance of GI plus the following standards: 2. Well-being standards. 3. Water standards. 4. Wildlife standards. Scoring: 1. Address/meet—a–e for core standards and a–c for well-being, water and wildlife = achieved. 2. Address/meet—a–e for core standards and a–c for well-being, water and wildlife + at least six from d–e for wellbeing, water and wildlife (nine indictors to work with) = excellent.
Woodland Access Standard	Woodland Trust	Woodland of 2 ha within 500 m of homes and 20 ha within 4 km of homes
Urban Greening Factor	Greater London Authority	Measurement of all different landcovers (each with a factor calculation for greenness/ecological function) divided by total area. Developments should achieve a score of 0.3–0.4 for onsite greening or above to be approved.
By all reasonable means: Least restrictive access to the outdoors	Natural Resources Wales	Based on four categories of access: 1. Least Restrictive Access—a principle applied to all work to ensure it aspires to the highest standards possible. 2. Access for All standards and By All Reasonable Means zones—the most widely adopted advisory standards and zoning approach—along with statutory standards relating to building design. 3. Access Chain—a tool that uses the steps of a visit to guide decisions about access improvements. 4. Combining access with quality of experience—a principle that ensures all access improvements match priorities for visitor experience.
Open Space Categorisation—London Plan (2016)	Mayor of London	Based on size (ha)/distance (km from home) focused on seven categories: 1. Regional parks (400 ha and 3.2–8 km). 2. Metropolitan parks (60 ha and up to 3.2 km). 3. District parks (20 ha and up to 1.2 km). 4. Local parks and open spaces (2 ha and within 400 m). 5. Small open spaces (<2 ha and <400 m). 6. Pocket parks (<0.4 ha and <40 0m). 7. Linear open space (variable/variable).

## Data Availability

Not applicable.

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
