# Peer review of "Access to Nature in a Post Covid-19 World: Opportunities for Green Infrastructure Financing, Distribution and Equitability in Urban Planning"

_ijerph, 2021, doi:10.3390/ijerph18041527_

Round 1
Reviewer 1 Report
Which is the focus of the paper? Is it the general issue of functionality and location of green infrastructure in urban areas or the theme of disparity in access linked to ethnic diversity, etc.? Or both? In any case, both issues appear barely sketched and unclear and yhey need to be focused. Some data, measures, indicators are simply cited using the bibliographic reference, when it would be necessary to make that reference explicit, expanding the citation and making it comparable with the other data. For example, this sentence:
"Moreover, Public Health England (2020) argued that green infrastructure was a crucial component of the UK’s response to Covid-19, and to improving public health more generally. Evidence released underpinning their Improving access to greenspace: A new review for 2020 report suggested that savings of £2.1 billion per year could be made in health costs if everyone in England had access to high-quality greenspace"
refers to a study that cannot be simply cited, but the methodology and the data and indicators used must be explained in the paper. In the same way, for the table 1 the connection with the general themes of the paper is not clear.
Finally, the theme of green infrastructure, its project and the policies to implement it, is clearly linked to urban and landscape planning and their tools (urban plans and projects). This aspect, despite the historical references in the introduction (Garden cities, the projects of Olmsted, etc.), is neglected in the paper, just as the themes of landscape urbanism and ecological urbanism appear to be missing. In general, the paper offer a contribution in the field of urban policies, but not in the field of urban and landscape planning and project, missing the focus on the shape, features and tipolgies of settlments and open spaces.
Author Response
We would like to thank all three reviewers for their insightful comments. Following a process of consideration we have made significant changes to the manuscript. Below we outline our specific responses to comments received on the initial submission.
Reviewer 1:
Which is the focus of the paper? Is it the general issue of functionality and location of green infrastructure in urban areas or the theme of disparity in access linked to ethnic diversity, etc.? Or both? In any case, both issues appear barely sketched and unclear and yhey need to be focused.
We have modified the introductory section and the framing to improve the clarity of the paper. We have made it clearer that the paper use issues of distribution and access to examine how these impact on use and by association health. We add in financial issues to situate the ongoing issues of access and equitability in terms of provision. This highlights within the latter sections the problems associated with access/equitable distribution in different communities to support our argument that although GI has benefits that are meaningful to individuals and society more widely that some people are being excluded or marginalised from benefitting from them. We have also extended/modified sections 3, 4 and 5 to make our argument and choice of areas of focus stronger.
Some data, measures, indicators are simply cited using the bibliographic reference, when it would be necessary to make that reference explicit, expanding the citation and making it comparable with the other data.
Thank you for the comment and we can appreciate the need for clarity. Where we have used specific data it is contextualised within a specific location, most frequently the UK, providing a continuity to the framing/evaluation of the data. Also we use a small range of data types – percentages of population, monetary values (£ and % losses), and proportional changes in size (m2/%). Although the paper is centred predominately on the UK it would not be valuable to add further detail of specific areas (total populations, total economic spending, total houses/businesses/proportions of GI), as this would be too discreet to show a wider picture. Moreover, although the impacts of austerity have been felt across the UK the ways in which this has impacted upon LPA funding differs depending on land holding, economic viability and opportunities, and historical urban form. To break down each form of data and provide this information would not, in our opinion, add to the paper. Moreover, this information can be found by reviewing the papers/reports cited in the text.
For example, this sentence:
"Moreover, Public Health England (2020) argued that green infrastructure was a crucial component of the UK’s response to Covid-19, and to improving public health more generally. Evidence released underpinning their Improving access to greenspace: A new review for 2020 report suggested that savings of £2.1 billion per year could be made in health costs if everyone in England had access to high-quality greenspace"
refers to a study that cannot be simply cited, but the methodology and the data and indicators used must be explained in the paper. In the same way, for the table 1 the connection with the general themes of the paper is not clear.
Whilst we understand and appreciate the reviewers comment we do not agree that we ned to provide a more detailed analysis of the methodologies and/or approaches used to each of the sources of information we cited. To do so would mean the paper would need to expand its scope and length to assess how each study was conducted. We would argue that Public Health England are a body with a strong reputation for developing health policy that is grounded in robust evidence, as they funded by the UK government. Therefore we do not feel that we need to interrogate their data in the text. If readers of the paper wish to do this they have the full bibliographic reference to do so. Moreover, within the GI, urban planning and health literature there is a consistent use of data and economic analysis to support argumentation that does not delve into specific methodologies or analytical techniques. Whilst other papers systematically do this kind of examination the purpose of this paper is not to assess who these figures were generated but to indicate that the work/research has been conducted and that there is a growing evidence base reporting the positives to public health of investing in GI.
We have added further information outlining the value/function of Table 1 to illustrate why an appreciation of types of GI, scale, funding bodies, and long-term viability of funding is important. In addition the paper makes reference to the different scales of GI and the ways in which value/meaning/benefits can be attributed to different types of space. We also make reference to the cumulative value of GI at the local/city scale which can be linked to Table 1. It is also important to highlight the growing variability of funding options during a period of austerity government which supports the use of Table 1 in the paper.
Finally, the theme of green infrastructure, its project and the policies to implement it, is clearly linked to urban and landscape planning and their tools (urban plans and projects). This aspect, despite the historical references in the introduction (Garden cities, the projects of Olmsted, etc.), is neglected in the paper, just as the themes of landscape urbanism and ecological urbanism appear to be missing. In general, the paper offer a contribution in the field of urban policies, but not in the field of urban and landscape planning and project, missing the focus on the shape, features and tipolgies of settlments and open spaces.
We have added only a brief introduction to GI as a theme/concept as these discussions have been held elsewhere. Moreover, we locate our thinking in the literature to set out the key principles/ideas we are working with within the paper (see for example references 17-20). We have not gone into further detail, as we do not believe it would add significantly to the paper and these discussions can be found elsewhere (they are signposted in references: 11, 15., 16, 17, 18, 19, 20, 21, 28, 31, 42, 45). We do though believe that briefly mentioning Olmsted and Howard is important as both reflected on the role of greenspace/GI on health which is a starting point for our discussion. The paper is also not focussed on assessing the role of plans, projects or tools to assess accessibility, equitability of distribution and financing. Alternatively, it provide a commentary of how these themes/areas of planning can be used to assess the ways in which GI can and is being used in urban areas. More detailed, and policy/project specific, evaluations are used to support our arguments, i.e. references 20, 29, 31, but this was not the aim of the paper. We have also not used aspects of landscape or ecological urbanism in the paper and in our opinion these are not as prominently discussed in the wider literature or used to examine GI policy/practice in the UK. Although these concepts have value they have not been added in this paper as we feel it would not enhance the paper significantly in terms of its grounding. The same is true for our lack of explicit debate of “shape, features and typologies of settlements and open spaces”, which we do not go into. If we were to provide this level of detail for each example we use the paper would become far larger and less focussed. Whilst this detail is useful and would have been provided if we had been working with a single case study for a wider commentary it does not offer significant additionality to the paper. Moreover, the choice of academic references, for example: 6, 15, 32, 41, 46, 54, and 55, provide locations where this additional detail can found.
We would though like to thank the reviewer for this comment as we feel it is important to reflect on issues of size, typology, location and geographical context. These issues have formed part of the analysis of references we cite, and this paper aims to provide a commentary signposting readers to these more specific case studies.
Reviewer 2 Report
The paper is a very well evidenced commentary on the opportunities for Green Infrastructure financing, distribution and equitability in urban planning situating green infrastructure in the Covid-19 debates. I believe it is a worth publishing paper as it provides a very good understanding of the value of Green Infrastructure bringing it at the forefront of the debates regarding urban planning in the post-Covid period.
My main comments and recommendations for minor revision are the following:
- I think that the authors should distinguish a little better when they talk on a wider international scale and when they refer to the UK case where, anyway, they draw their main evidence from and upon which they build their comments and specify their recommendations. For example, section 3 has a title “Financing green infrastructure in the UK” (thus stating that it talks only about the UK case) while sections 4 (Equitability) and 5 (Distribution) apparently raise also issues drawing form other international cases. This could be simply corrected by deleting the UK from the title in section 3 and stating at the beginning of the section “Drawing for the UK case etc.)
- I don’t find some of the notes so necessary. For example, note 4 could be included in the main text, others such as note 3 are not really necessary in a commentary paper.
- A small point for the Distribution parameter. Smaller spaces that service neighborhoods can be of equal importance as a city-scale park not only in terms of accessibility or functionality (lines 278-279) but also regarding their cooling effect and therefore regarding adaptation to climate change, especially in compactly built-up areas (see for example, Yiannakou, A., & Salata, K.D. (2017). Adaptation to climate change through Spatial Planning in Compact Urban Areas: A Case Study in the City of Thessaloniki. Sustainability, 9(2), 271. https://www.mdpi.com/2071-1050/9/2/271)
Author Response
Reviewer 2:
The paper is a very well evidenced commentary on the opportunities for Green Infrastructure financing, distribution and equitability in urban planning situating green infrastructure in the Covid-19 debates. I believe it is a worth publishing paper as it provides a very good understanding of the value of Green Infrastructure bringing it at the forefront of the debates regarding urban planning in the post-Covid period.
My main comments and recommendations for minor revision are the following:
- I think that the authors should distinguish a little better when they talk on a wider international scale and when they refer to the UK case where, anyway, they draw their main evidence from and upon which they build their comments and specify their recommendations. For example, section 3 has a title “Financing green infrastructure in the UK” (thus stating that it talks only about the UK case) while sections 4 (Equitability) and 5 (Distribution) apparently raise also issues drawing form other international cases. This could be simply corrected by deleting the UK from the title in section 3 and stating at the beginning of the section “Drawing for the UK case etc.)
- We have added the following sentence into the introduction to help clarify this situation:
“We do this from a UK perspective using examples of policy, practice and financing from this geographical location. However, this is supplemented with additional commentary from outside of the UK to illustrate that these are not isolated issues only relevant to the UK.”
We have also made sure that within the text we do not shift geographical location too often, and that where this occurs we relate the non-UK discussions to UK practice/actions. In Sections 4 and 5 we have made reference to the added-value of engaging with international debates to illustrate the complexity of approaches/debates taking place in the UK.
- I don’t find some of the notes so necessary. For example, note 4 could be included in the main text, others such as note 3 are not really necessary in a commentary paper.
- We have removed the footnotes and incorporated them where they add to the narrative and removed them in other places.
- A small point for the Distribution parameter. Smaller spaces that service neighborhoods can be of equal importance as a city-scale park not only in terms of accessibility or functionality (lines 278-279) but also regarding their cooling effect and therefore regarding adaptation to climate change, especially in compactly built-up areas (see for example, Yiannakou, A., & Salata, K.D. (2017). Adaptation to climate change through Spatial Planning in Compact Urban Areas: A Case Study in the City of Thessaloniki. Sustainability, 9(2), 271. https://www.mdpi.com/2071-1050/9/2/271)
- Thank you for this comment. We have amended the paragraph to include this point and the reference suggested.
Reviewer 3 Report
Type of manuscript: Commentary
Title: Access to nature post Covid-19: opportunities for green infrastructure
financing, distribution and equitability in urban planning
A brief summary (one short paragraph) outlining the aim of the paper and its main contributions.
The aim of the paper is to discuss the role of green infrastructure (GI) for health in urban/environmental planning, in the light of pandemic, and to propose measures to better address long-term underfunding of GI which have had effects on distribution, equitability and access of GI, and especially so for poorer (urban) communities.
Broad comments highlighting areas of strength and weakness. These comments should be specific enough for authors to be able to respond.
The focus of the paper is very timely. The intersection between greenery and health has come to the fore in an interesting way during the pandemic. There is clear need to discuss the past and future pathways and priorities with regards to how GI is dealt with in urban planning. Overall, the paper is well written and structured, brings up many important aspects for discussions, and also suggests ways to move forward. A reflection: The suggestions made by the authors addresses top-down strategies and structures. Is there also a role for “public participation”, protests,citizen science and the like to ensure and protect GI in urban development?
Specific comments referring to line numbers, tables or figures. Reviewers need not comment on formatting issues that do not obscure the meaning of the paper, as these will be addressed by editors.
Comments and suggestions on Aim and scope
The paper lacks a clear aim.
- I suggest that authors add a more explicit aim, especially when it comes to the focus on funding/financing, and perhaps one or two questions that can help guide the structure of the papers and limits it scope.
After reading the introduction twice, I understand that the pandemic it works more as a backdrop for discussing important issues regarding GI. However, with a title involving Covid, and the importance of health issues stressed in the introduction, makes the reader expect an ever stronger focus on the role of GI from a health perspective and for mediating the consequences of the pandemic and associated restrictions on mobility and the need for keeping distance.
- I suggest some comments on what the reader can expect in terms of Covid/pandemic lenses on the GI in an outline presented in the Introduction-section.
On line 114-116 is there problem-formulation addressing the role of funding, it reads: “The lack of an baseline economic position for green infrastructure has therefore remained problematic raising concerns within government of whether nature is cost-effective or indeed needed to support a prosperous society (Young, 2010).”
- I suggest that the above formulation, and the section it belongs to is introduced early in the paper, in the Introduction section, which can help formulating a clearer aim and scope.
- A general suggestions is to drop the footnotes, integrate some of them in the text, or at least reduce their numbers.
Line 53-54 are there references to “Lennon, 2020; Rastandeh and Jarchow, 2020 referred”, two papers that to more explicitly deal with GI in the face of current pandemic.
- I suggest that the authors provides some more details of the main arguments in these two articles, as it might clarify the contribution by this Commentary.
Comments and suggestions on Table 1
Table 1 (line 198) is interesting in how it dissects GI into different types of spaces and scales. However, the table needs some further explanation and integration into the text to fully make sense. Does the table reflects the UK in total, or does it draw from Liverpool (a sentence on line 179 could suggest that)?
- I suggest that table text (line 199-205) is inserted in the main text and that the authors explain better how the table came about.
Further reflections on Table 1: Can the authors detect a trend towards more “effective”, small scale promotion of GI, on the behalf of funding more extensive green space areas? And further, it could also be discussed if lack of funding necessarily is a problem for access and attractiveness of green space, as there might be recreation qualities in green areas partially “forgotten” (unintentionally “re-wilded”).
Further comments and suggestions
Reflection on line 278-279, arguments about the role of scale of GI (also see comment above). Small greens space in the vicinity of people’s homes are important, pandemics or nor. But from an outdoor recreation perspective, has not the importance of large scale GI become even more important, e.g. for keeping distance, reducing wear and tear?
I find Section 4 Equitability and 5. Distribution is interesting and pressing, and well aligned with the papers focus on the consequences of the pandemic. A note on headlining: “Equitability” (in what regard?) and “Distribution” (of what?) are short, and should be elaborated in order to better capture the content of the respective sections.
Section “6.4 Establishing an economic value for green infrastructure”. I think that the argument for economic valuation could be strengthened by comment on the most common ways of (economic) valuation. Without being an expert in this area, one might expect that e.g. contingent valuation (e.g. WTA and WTP) and valuations that build on adjacent land-price is likely to devalue GI located in poorer neighborhoods.
- I suggest that you elaborate a little bit more about how economic valuation might help address issues associated with equitability and distribution addressed previously in the paper.
- I finally, if there is space, I suggest a “6.5 Suggestions for further research” .
Author Response
A brief summary (one short paragraph) outlining the aim of the paper and its main contributions.
The aim of the paper is to discuss the role of green infrastructure (GI) for health in urban/environmental planning, in the light of pandemic, and to propose measures to better address long-term underfunding of GI which have had effects on distribution, equitability and access of GI, and especially so for poorer (urban) communities.
Broad comments highlighting areas of strength and weakness. These comments should be specific enough for authors to be able to respond.
The focus of the paper is very timely. The intersection between greenery and health has come to the fore in an interesting way during the pandemic. There is clear need to discuss the past and future pathways and priorities with regards to how GI is dealt with in urban planning. Overall, the paper is well written and structured, brings up many important aspects for discussions, and also suggests ways to move forward. A reflection: The suggestions made by the authors addresses top-down strategies and structures. Is there also a role for “public participation”, protests, citizen science and the like to ensure and protect GI in urban development?
We have added a small number of comments to highlight the important role of local communities in these processes and the current variability in their engagement/use by policy and decision-makers. This provides an additional facet to the discussion that moves beyond LPAs/developers and their contribution to addressing equity and distribution issues.
Specific comments referring to line numbers, tables or figures. Reviewers need not comment on formatting issues that do not obscure the meaning of the paper, as these will be addressed by editors.
Comments and suggestions on Aim and scope
The paper lacks a clear aim.
- I suggest that authors add a more explicit aim, especially when it comes to the focus on funding/financing, and perhaps one or two questions that can help guide the structure of the papers and limits it scope.
Thank you for this comment. We agree with the reviewer and have added additional information that sets out a problem and what the paper will and will not do in terms of its commentary (rather than new empirical focus).
After reading the introduction twice, I understand that the pandemic it works more as a backdrop for discussing important issues regarding GI. However, with a title involving Covid, and the importance of health issues stressed in the introduction, makes the reader expect an ever stronger focus on the role of GI from a health perspective and for mediating the consequences of the pandemic and associated restrictions on mobility and the need for keeping distance.
- I suggest some comments on what the reader can expect in terms of Covid/pandemic lenses on the GI in an outline presented in the Introduction-section.
We have added further commentary to clarify what the paper does and do not do. This includes a statement outlining our discussions of equitability, distribution and finance as important factors in understanding responses to Covid-19 rather discussing health explicitly. We do though make reference to recent work looking at GI/Covid-19/health.
On line 114-116 is there problem-formulation addressing the role of funding, it reads: “The lack of an baseline economic position for green infrastructure has therefore remained problematic raising concerns within government of whether nature is cost-effective or indeed needed to support a prosperous society (Young, 2010).”
- I suggest that the above formulation, and the section it belongs to is introduced early in the paper, in the Introduction section, which can help formulating a clearer aim and scope.
- A general suggestions is to drop the footnotes, integrate some of them in the text, or at least reduce their numbers.
We have removed all footnotes except one and incorporated the comments into the text. The remaining footnote has been changed into an endnote. We have done this as we felt that the discussion of lockdown tiers was important but not crucial for the in-text discussion. We have also moved the statement using the Young (2010) material into the introduction to support the need for an discussion of economic issues within a broader discussion of equitability and distribution.
Line 53-54 are there references to “Lennon, 2020; Rastandeh and Jarchow, 2020 referred”, two papers that to more explicitly deal with GI in the face of current pandemic.
- I suggest that the authors provides some more details of the main arguments in these two articles, as it might clarify the contribution by this Commentary.
We have amended this sentence and added in further details regarding what both papers proposed and linked it to the focus of this paper. This has been aligned with the modifications made in the introduction to improve the clarity of what our paper does and how it builds/uses these discussions to situate the arguments being made.
Comments and suggestions on Table 1
Table 1 (line 198) is interesting in how it dissects GI into different types of spaces and scales. However, the table needs some further explanation and integration into the text to fully make sense. Does the table reflects the UK in total, or does it draw from Liverpool (a sentence on line 179 could suggest that)?
- I suggest that table text (line 199-205) is inserted in the main text and that the authors explain better how the table came about.
We have added the supporting/explanation text before Table 1 as suggested.
Further reflections on Table 1: Can the authors detect a trend towards more “effective”, small scale promotion of GI, on the behalf of funding more extensive green space areas? And further, it could also be discussed if lack of funding necessarily is a problem for access and attractiveness of green space, as there might be recreation qualities in green areas partially “forgotten” (unintentionally “re-wilded”).
Thank you for this comment. We agree and have added additional commentary regarding issues of scale with local/discreet being funding more than larger sites. We have also made reference to two smaller approaches guerrilla gardening and alleyway greening as examples of community led approaches that are outside of formal funding mechanisms.
Further comments and suggestions
Reflection on line 278-279, arguments about the role of scale of GI (also see comment above). Small greens space in the vicinity of people’s homes are important, pandemics or nor. But from an outdoor recreation perspective, has not the importance of large scale GI become even more important, e.g. for keeping distance, reducing wear and tear?
We have added in an additional set of statements discussing this point and noting that larger sites have a greater capacity to support users and uses, and that we need to consider GI as a set of different sized spaces that are part of a multi-functional network.
I find Section 4 Equitability and 5. Distribution is interesting and pressing, and well aligned with the papers focus on the consequences of the pandemic. A note on headlining: “Equitability” (in what regard?) and “Distribution” (of what?) are short, and should be elaborated in order to better capture the content of the respective sections.
Thank you for this comment. We have elongated the titles to make them more directed. Equitability relates to access to GI and Distribution relates to the spatial locations/layout of GI in urban areas.
Section “6.4 Establishing an economic value for green infrastructure”. I think that the argument for economic valuation could be strengthened by comment on the most common ways of (economic) valuation. Without being an expert in this area, one might expect that e.g. contingent valuation (e.g. WTA and WTP) and valuations that build on adjacent land-price is likely to devalue GI located in poorer neighborhoods.
- I suggest that you elaborate a little bit more about how economic valuation might help address issues associated with equitability and distribution addressed previously in the paper.
- We have added additional text to 6.4 to signpost the use of different valuation techniques and to raise the concern regarding valuation that is detrimental to other spaces in a given location.
- I finally, if there is space, I suggest a “6.5 Suggestions for further research” .
- Due to the constraint of the word limit we have not explicitly added a new section outlining areas for further research. We have however made reference in the text to the need to do further work to assess the links between equitability, distribution, access and funding.
Round 2
Reviewer 1 Report
The clarity of the new paper, in particular of the focus, has certainly been improved.
The reviewer agrees with many of the contents of the authors' reply, which clarify some issues of the paper.
However, specifying better what we are talking about when talking about GI is useful to better frame the focus, not to make it vaguer, as the authors claim. Therefore, I would like to recommend improving the concept of green (and perhaps blue…) infrastructures (also in term of shapes, features and typologies of settlements and open spaces) also with some EU documents, given the context covered by the paper (i.e., EC (2019), Additional information on the review of implementation of the EU green infrastructure strategy. 2019/184 final; or https://ec.europa.eu/environment/nature/ecosystems/index_en.htm; etc.).
Author Response
We would like to thank the reviewer for their comments.
We wrestled with how much discussion of what GI is, what it does, and what form it should it to add to the paper. We decided initially to minimise this to provide scope for the discussion of accessibility, functionality and financing to take prominence. However, after receiving this comment we have amended the paper and added two additional paragraphs into page 2 to highlight the variation in understanding. This is supported by a new Table 1 (the existing Table 1 and 2 have been renamed Table 2 and 3) which outlines the variety of types, scales, and benefits associated with GI and whether these elements are a site based GI or part of a corridor/network perspective.
We have also used the existing literature quoted in the paper to support this discussion. We acknowledge that there is a wealth discussion reflecting on what GI is and the different typologies that have been created. However, although this work is useful, we do not feel it adds sufficiently to the arguments being made. We have therefore not delved too much into the typology discussions produced by the EU, Natural England in the UK, EPA in the USA or others, as we do not feel that this would added significantly to the paper.
We do agree that this is an interesting area of investigation and we will look at how we might address issues of typologies and access/functionality in subsequent work.